# Single-Cell and Single-Nucleus RNAseq Analysis of Adult Neurogenesis

**DOI:** 10.3390/cells11101633

**Published:** 2022-05-13

**Authors:** Alena Kalinina, Diane Lagace

**Affiliations:** Neuroscience Program, Department of Cellular and Molecular Medicine, Ottawa Hospital Research Institute, Brain and Mind Research Institute, University of Ottawa, Ottawa, ON K1H 8M5, Canada; akali103@uottawa.ca

**Keywords:** adult neurogenesis, single-cell/single-nucleus RNA sequencing, single-nucleus, subventricular zone, dentate gyrus, hypothalamus

## Abstract

The complexity of adult neurogenesis is becoming increasingly apparent as we learn more about cellular heterogeneity and diversity of the neurogenic lineages and stem cell niches within the adult brain. This complexity has been unraveled in part due to single-cell and single-nucleus RNA sequencing (sc-RNAseq and sn-RNAseq) studies that have focused on adult neurogenesis. This review summarizes 33 published studies in the field of adult neurogenesis that have used sc- or sn-RNAseq methods to answer questions about the three main regions that host adult neural stem cells (NSCs): the subventricular zone (SVZ), the dentate gyrus (DG) of the hippocampus, and the hypothalamus. The review explores the similarities and differences in methodology between these studies and provides an overview of how these studies have advanced the field and expanded possibilities for the future.

Single-cell and single-nucleus RNA sequencing (sc-RNAseq and sn-RNAseq) studies have provided unparalleled insight into the transcriptional programs of different cellular states during the process of adult neurogenesis by measuring the transcriptomes of thousands of individual cells. This review summarizes 33 studies examining adult neurogenesis that have either created or used sc- or sn-RNAseq datasets from the subventricular zone (SVZ) (Table 1, *n* = 17), dentate gyrus (DG) of the hippocampus (Table 2, *n* = 17), and the hypothalamus [1,2,3]. Given the larger number of studies performed in the SVZ and DG, the review first summarizes the methodological variations between these studies in terms of the species and types of models used; differences in dissection, isolation, purification of the single cells and nuclei protocols; as well as the variety of platforms and analysis pipelines utilized. This is followed by a review of the number and type of cells identified, the transcriptional dynamics within the SVZ, DG, and hypothalamus, as well as insights gained on neurogenesis within the aging and injured brain.

## 1. Methodology: Species Demographics

Mice were used in all the reviewed studies, with the exception of four publications on the hippocampus. Cynomolgus macaques were used by Zhang et al. [31] in 2021 and provided the first snRNA analysis of frozen post-mortem hippocampus samples from eight young (four-to-six-years old) and eight aged (18–21 years old) primate animals. Sn-RNA analysis was first published in human samples in 2017 by Habib et al. [24], followed in 2021 by Tran et al. [33] and Ayhan et al. [34], and most recently in 2022 by Franjic et al. [36]. The work by Habib et al. [24] utilized human samples from four non-diseased donors aged 40–65 that were obtained from the Genotype-Tissue Expression (GTEx) project. This dataset was also reanalyzed by Sorrels et al. in 2021 [35]. The work by Tran et al. used the hippocampus from three neurotypical donors with an age range of 40–69 [33], which is in contrast to Ayhan et al. [34], who isolated the hippocampus from five patients aged 24–60 that were undergoing surgical treatment for epilepsy. In 2022, the first examination of the neurogenic lineage cells in the adult human hippocampus-entorhinal system was performed by Franjic et al. [36]. This study included six clinically unremarkable human donors with a mean age of 52 years old and used sn-RNAseq analysis in the adult rhesus macaques, young adult pigs, and previously published mouse data [26] that reveal species-specific differences.

In addition to the comparison done in young and old cynomolgus macaques [31], within the mouse literature there are four studies that have compared different ages directly (*n* = 3, SVZ, Table 1; *n* = 2, DG Table 2). This work has identified aging-specific differences, for example, when analyzing mice across different life stages at two weeks, two, six, and 12 months [17], three and 28–29 months [12], in two, 22–23 month-old animals [10], and in two and 14-month-old animals [37]. Aside from these few studies that examined more than one age, the vast majority of studies, in both the SVZ and SGZ have focused on relatively young mice of less than 12 weeks of age, which affects the ability to make generalizations about the findings obtained.

Between the studies there is also diversity in terms of whether both sexes were used, and how sex-based differences were tested. More specifically, the datasets have included either the use of male or female samples, combined males and females into a pooled sample, kept the two sexes unpooled, or pooled the sexes and used Multi-seq barcodes to identify the males and females. For the SVZ datasets, 45% were created using male mice, and the remaining studies used either female mice (*n* = 1), both sexes (*n* = 3), both sexes unpooled (*n* = 2), or multiplexed pooled samples (*n* = 1). Of the studies that analyzed unpooled samples, only one reported sexual dimorphism, which showed that male mice have a higher number of oligodendrocyte progenitors within the septal SVZ wall [11]. In contrast, a later study that multiplexed pooled samples with high success did not identify sexual dimorphisms in the neurogenic transcriptome within the SVZ [19]. The hippocampal datasets generated from mice included male mice (*n* = 3), female mice (*n* = 1), both sexes pooled (*n* = 4), or both sexes pooled but separated later for downstream analyses (*n* = 1). The study performing downstream sex-based analyses in the mouse DG examined the sum expression of sex-specific genes like Ddx3y and Xist [26]; however, sex-based differences in adult neurogenesis could not be adequately addressed using this method. Similarly, the hippocampal datasets generated from primates included males [24,33], as well as females and males separately [31,34,36], with one study citing direct comparisons [31]. Male and female cynomolgus monkeys showed no sex differences in the cell distributions, but suggested that the male hippocampus may be more susceptible to aging based on a larger number of differentially expressed genes in males compared to female groups across almost all cell types [31]. Given the successful use of multiplexed pooled female and male samples, it is only a matter of time before other researchers use this technique with high success in the DG. Furthermore, it will be important for researchers, reviewers, and editors to continue to ensure the sex of the animals used and the method for combining samples and any sex-based analysis is published to increase the rigor and reproducibility of this work.

## 2. Methodology: Sample Collection

An approximately equal number of studies have been published using samples collected from the SVZ (Table 1, *n* = 17) and the DG (Table 2, *n* = 17). The open-source availability of the high-throughput single-cell datasets associated with these papers have paved the way for other researchers to use these datasets to enrich their own analyses. Datasets have often been used in subsequent studies to obtain new results based on testing new hypotheses, or to compare results for consistency and cross-referencing. For example, the most commonly used SVZ dataset comes from the early work by Llorens-Bobadilla et al. [4], which has been subsequently utilized in five studies. Similarly, the seminal work of Hochgerner et al. [26] in the DG has been subsequently utilized in six studies. In addition, seven studies (*n* = 2 in SVZ; *n* = 5 in DG) did not create new datasets but performed an additional analysis, and/or used already published datasets to refine methodology. Included in this group is the work by Sorrells et al. [35], who reanalyzed the neural stem cells (NSCs) identified in the human postmortem DG sample by Habib et al. [24]. Given the continuous development of new analysis pipelines, it seems likely that the use of published datasets will continue to become more prevalent in the future.

The areas of microdissection for either the SVZ or hippocampus have varied vastly between the studies. For the dissection of the SVZ, the vast majority of papers (*n* = 9) dissected the lateral wall of the ventricle (Table 1). However, over time there is a trend for different subregions of the SVZ to be examined. This includes analysis of the septal and lateral SVZ [11], the SVZ and corresponding OB [13], the dorsal and ventral region [15], and, most recently, examination of the anterior-dorsal, posterior-dorsal, anterior-ventral, and posterior-ventral regions [19]. For dissection of the mouse hippocampus, there are three seminal studies that used microdissected DG and two studies that used hippocampal anatomical subregions (DG, CA1, CA2, and CA3) [6,19]. Within the paper examining primates, three studies included the hippocampus [24,31,33], whereas Ayhan et al. [34] used the anterior and posterior hippocampus, and Franjic et al. [36] included five microdissected subregions (SGZ, CA2-CA4), CA1, Sub, and EC). This variability in dissected regions in both the SVZ and hippocampus thus allows for a rich number of datasets for future data mining, and reveals spatial differences in cell types residing within the SVZ and DG.

A critical component of the protocol for sc- and sn-RNAseq is the isolation of high-fidelity single cells or nuclei and the generation of single-cell suspensions. The use of sc-RNAseq compared to sn-RNAseq in adult brain studies appears to provide better results, capturing more transcripts per cell, as well as resulting in a more successful mapping of the transcriptome [22,38,39]. However, some cells require freezing, fixation, staining, or harsher and more extended treatments that may impact their viability or transcriptomic integrity, which supports the use of sn-RNAseq [3]. Likely due to the fairly short procedure for dissociation of live cells from the brain, and later development of sn-RNAseq, studies have utilized sc-RNAseq more extensively than sn-RNAseq in the SVZ and DG. Indeed, only two studies have isolated single nuclei in the SVZ, and six studies in the DG. Included in this are all five of the datasets generated from primates [24,31,33,34,36]. The work with human samples had a notably large range in postmortem intervals (PMI). The shortest PMI was 12 min, with the samples being removed from patients with epilepsy undergoing surgery treatment using an en bloc resection technique that allowed the hippocampus to be dissected from its vascular pedicle immediately prior to tissue processing [34]. This is in contrast to the others that included an average PMI of 12.5 h [24], 20–38 h [33]. and 9–12 h [36]. To control for the possible confounding effects of PMI, Franjic et al. [36] also subjected their young adult pig samples to 30 min, 1 h, and 7 h of warm ischemic PMI and found no effect of duration. However, questions remain whether this shorter time period in the pig was sufficient to recapitulate the possible effect that occurred in human samples that had a PMI of 9–12 h [40].

The collection, dissociation and storage methods of cells or nuclei from the brain also varied within the studies (Table 1 and Table 2). For example, within the SVZ, Zywitza et al. [8] used papain digestion and methanol fixation, Llorens-Bobadilla et al. [4] used trypsin digestion and FACS of cells that were subsequently frozen, and Mizrak et al. [11] processed the cells live. One paper also provided a comparison of datasets using papain versus trypsin for digestion, which revealed that the transcriptional dynamics of NSC regulators were very similar with respect to the expression of key dynamically regulated genes [5]. In contrast to this variability, the work in humans has commonly used either sucrose gradient centrifugation [24,36], or the “Frankenstein” method of nuclei isolation [24,33,34]. The latter method is highly utilized and includes the use of FACS to identify cell subpopulations based on ploidy to ensure the isolation of single nuclei and the removal of debris and ambient RNA to help reduce the background [23]. Given the diversity of methods utilized, researchers should be aware of biases and benefits of different dissociation and storage methods for cell preparation that may confound outcomes. Thus, publishing work that addresses this concern in the future, even as part of larger studies, will aid the field greatly.

After the generation of single-cell suspensions, 70% of studies have enriched and purified the samples to isolate the cell types of interest prior to analysis. Purification has been performed most often in both the SVZ and DG using isolation for fluorescence-tagged cells from reporter mice, conditional or inducible transgenic mice, or virus-injected mice. This is in comparison to fewer studies using immunofluorescent staining of live or fixed cell/nuclei suspensions. In the SVZ there are four datasets that used reporter mice, including two that used the GFAP promoter, and one that used both the Ki67 promoter and the CGD promoter. In contrast, in the DG there are four datasets that used reporter mice, with two using the Nestin promoter and two using the GFAP promoter. Conditional (*n* = 3) and inducible (*n* = 6) Cre transgenic mouse models with a variety of promoters have also been used in creating datasets in the SVZ, yet have not been used to create datasets in the DG. One of the strengths of the conditional or inducible mice is that these models label the cell type of interest and all their progeny, thus providing a robust way to track lineage, but sometimes make it more difficult to identify when temporal changes occur within cell populations. This limitation can be overcome through the use of more animals at varying time points, or by combining different mouse models. For example, Magnusson et al. [14] used the Cre-inducible transgenic model and AAV-Cre virus-mediated recombination to label different cell populations and compare the datasets. There are also concerns that treatment with tamoxifen in Cre-inducible mice may have confounding effects. This was in part addressed in a sc-RNAseq study by Lee et al. [41] which showed that prenatal treatment with tamoxifen alters cortical neurogenesis. This paper also found that in three to four week old adult mice, tamoxifen treatment reduces proliferation as measured by immunohistochemical methods, whereas others have not found such effects in mice at four to seven weeks [42] or five months of age [43] using similar methods. It is possible that these differences are due to strain-, age-, or dosage- dependent effects and thus the use of sc-RNAseq in the future could help address the long-lasting debate about the potential effect of tamoxifen on adult neurogenesis.

## 3. Methodology: Plate-versus Droplet-Based Methods and Number of Cells Obtained

While both plate-based and droplet-based methods are used in the SVZ and DG, droplet-based methods are chosen predominantly. Specifically in the SVZ and SGZ, there were 60% and 60% of studies that used droplet-based methods, respectively. For the plate-based methods, such as Smart-Seq2, individual cells are directly sorted into a well-plate either using a FACS procedure, with each well containing lysing reagents and barcoded primers bound to a microbead, or a fluorescence-based selection with a microscope. For the more commonly used droplet-based methods, such as the Chromium’s 10x Genomics, microfluidics are used to isolate each cell into a droplet containing the same reagents. The microbeads have unique well- or droplet-specific barcodes as well as unique molecular identifiers for each transcript. This procedure allows for precise tagging and identification of individual templates for each cell. The microbeads are isolated and pulled RNA is reverse-transcribed and amplified by PCR, followed by the addition of adapters for library preparation and sequencing. Both droplet- and plate-based sc-RNAseq protocols require the sequencing of obtained libraries, such as NextSeq or HiSeq platforms by Illumina.

The more common use of droplet-based methods within the field of adult neurogenesis is not surprising since it is well known to allow for a higher throughput study of single-cell transcriptomes and more resolution of the rare populations within heterogeneous niches, which is the case in the study of adult neurogenesis. Indeed, as summarized in Table 1 and Table 2, droplet-based methods are allowed to capture transcripts of several thousands to tens of thousands of cells e.g., [5,7,19], whereas plate-based methods rarely reached a quantity above a few thousand cells e.g., [9,20,21]. In addition, the multiplexing of samples, which has been noted to be successful to identify male and female mice in SVZ samples [19], also provides the benefit of more comparisons to be made while reducing the cost and number of animals required.

## 4. Methodology: Analysis Pipelines

Once the data matrices are obtained, machine learning algorithms are used to analyze the information in a high-dimensional space. A large variety of pipelines has been used to manipulate sc-RNAseq data for clustering by cell type and to analyze differential gene expression, pathway dynamics, lineage tracing, pseudotemporal resolution, and RNA splicing. In general, the most common platforms for cell clustering and differential gene expression are R-based, such as Seurat developed by Satija lab [44] and Monocle developed by Trapnell lab [45]. In addition to these popular methods, python- and Matlab-based analyses have also been used. Specifically, in the SVZ, 2/3rd (*n* = 10) used Seurat and 1/3rd (*n* = 5) used Monocle. In the DG, ten used Seurat and three used Monocle, while others relied on a few other methods ranging from Waterfall to Matlab-based analyses. There are many studies that have combined a few pipelines to maximize the information that can be obtained from the dataset, which also allows for the verification of results across platforms and enhanced interpretation (Table 1 and Table 2). For instance, python-based scVelo [28] or Velocyto [46] are often used to analyze induction and repression of genes via RNA velocity estimation together with Seurat or Monocle’s Pseudotime for complete developmental trajectory inference. In fact, a whole array of tools is being developed to study cell fate and regulation of gene expression based on leveraging the information about spliced and unspliced RNA counts alone [30,32].

## 5. Variability in Methodology in Examination of SVZ and DG

Summarizing these methodological variations highlighted that the studies in both the SVZ and DG vastly differ in types of mice utilized, methodologies for dissection, isolation, purification of the single cells or nuclei, and can be used in either plate-based or droplet-based methods and analyzed via a wide array of pipelines. Although these differences in the utilized methodologies may seem somewhat overwhelming, they enhance the robustness of datasets to answer new questions in the field of adult neurogenesis. It is also striking that, despite this variability, there are many similarities in outcomes and take-home messages from this large body of work. Included in these similarities are the various cell types that are commonly identified in the dissected areas, pseudotemporal resolution of developmental trajectories of adult-born cells, and the differences between reciprocal cell types. Furthermore, these studies provide some common findings and challenges left to solve in regards to the dynamic process of adult neurogenesis in the SVZ and DG.

## 6. Identification of Cell Types within the SVZ

One significant development resulting from sc- and sn-RNAseq analyses in adult neurogenesis studies was the generation of cell atlases. Zywitza et al. [8] was the first to study unsorted live and fixed cells from whole SVZ and resolved 17 clusters of cells including endothelial cells, pericytes, smooth muscle cells, microglia, perivascular macrophages, ependymal cells, medium spiny neurons (two types), oligodendrocytes (four types), astrocytes, neuroblasts (early and late), transient amplifying neural progenitors (NPCs, mitotic and not), and NSCs. Using FAC-sorted cells, Dulken et al. [5] found similar cell types in their study, except for some differences in the resolution of oligodendrocytes, the absence of ependymal cells, and no distinction between astrocytes and quiescent NSCs (qNSCs). The absence of ependymal cells may be due to their similarity to NSCs, which were subsequently found to be distinct using sc-RNAseq analysis of the SMACre^ERT2^ ependymal inducible mouse model [9]. Chen et al. [18] used homogenized frozen tissue chunks and identified 19 instead of 17 clusters with similar resolutions to Zywitza et al. [8]. Specifically, they found all the same cell types except for macrophages, and got a larger number of clusters due to the higher resolution of neuronal subtypes. This included identifying early versus late activated NSCs (aNSCs), as well as better resolution between qNSCs and astrocytes. Thus, overall there has been a large consensus, with up to 19 distinct subtypes of cells residing in the adult SVZ.

Mizrak et al. [11,13] performed a deeper analysis of SVZ heterogeneity by examining the ventral and septal walls of the SVZ as sources of variation between different cell types and their frequencies. One of the most striking findings of this analysis is that the lineage residing in the ventral wall is more biased towards giving rise to cells of neuronal fate, and that the septal wall has a gliogenic bias. In addition, heterogeneity of ventral and septal astrocytes correlated with NPCs and oligodendrocyte progenitor cells (OPCs), respectively. They have identified several clusters of astrocytes, some of which were positive for qNSC markers like Id2 and Hopx, as well as transition cells positive for aNSC markers Ascl1 and Egfr, although the classification of qNSCs was not separate from astrocytes. This led to the seminal finding that regional differences between septal and ventral NSCs and astrocytes contribute to the phenotypical heterogeneity of the SVZ.

## 7. Identification of Cell Types within the Dentate

The most extensive characterization of the cell types residing within the dentate at different developmental stages has been performed using hGFAP^GFP^ reporter mice by Hochgerner et al. [26]. This dataset encompasses perinatal, juvenile, and adult animals, and analyzed 24,000 cells resulting in the identification of 23 cell clusters in the adult mice. This included a large number of different neuronal cell types including two types of granule neurons (mature and immature), two types of neuroblasts, NPCs, Cajal-Retizus cells, three types of excitatory mossy cells, and three types of inhibitory neurons. There was also a large number of different types of glial cells, including astrocytes, oligodendrocytes (mature, precursors, newly formed), microglia, and radial glia-like NSCs. Mature astrocytes separated from RG-like NSCs (also known as qNSCs), but the cycling RG NSCs (also known as aNSCs) had to be separated from NPCs manually. Lastly, the dataset identified a number of vascular-associated cells including endothelial cells, microglia, perivascular macrophages, as well as vascular leptomeningeal cells.

When comparing the hippocampus to SVZ, it is likely more difficult to isolate the relatively small populations of proliferating NSCs and NPCs given the overwhelming number of mature neurons. Artegiani et al. [25], therefore, took the approach of purifying all non-neuronal cell types in the DG by negative sorting (GluR1-/CD24-), arguably to reduce the selection bias of the experiment. This resulted in the identification of 11 clusters of non-neuronal cells, including microglia, pericytes, interneurons, oligodendrocyte precursor cells, myelin-forming oligodendrocyte cells, endothelial cells, NPCs and NSCs. This study also showed that NSCs and NPCs exist on a continuum of states in mice, which can be seen using Pseudotime and examining the expression of quiescence, cell cycle, and neuronal specification genes.

In the human hippocampus, the use of sn-RNAseq to identify adult-generated NSCs and NPCs has suggested that very few, if any, exist in the adult. Habib et al. [24] first observed a cluster of 201 cells that were identified as NSCs in the human hippocampus based on putative NSC marker genes. Ayhan et al. [34] later demonstrated that the human hippocampus did not have glial cells with a stem cell identity, which was inconsistent with Habib et al. [24]. Similarly, Tran et al. [33] only observed clusters of astrocytes in the DG. Sorrells et al. [35] performed a reanalysis of Habib et al.’s [24] dataset combined with an enrichment analysis of ependymal markers and identified that the labelled NSCs in Habib et al. [24] were ependymal cells. In addition, they analyzed DCX expression in the same dataset and found scattered DCX expression in various cell types, with only 1.1% of the hippocampal cells expressing extremely low levels. These data led the authors to conclude that if any neurogenesis continues in the adult human DG, it is a rare phenomenon. In 2022, Franjic et al. [36] used Seurat and Monocle integration of all DG cells, but also performed a reintegrated analysis with only the granule cell lineage across the mouse, pig, macaque, and human. This reintegration analysis allowed for the identification of five clusters of cells including astrocytes/RG-like qNSCs, activated RGL cells (aNSCs), neural intermediate progenitor cells (NPCs), neuroblasts and granule cells. RNA velocity further provided progenitor and neuroblast trajectories for adult neurogenesis in the mouse, pig and macaque, in the absence of any clear trajectory in humans. In the human, a total of 20 cells were found to have an identity resembling NPCs, and only two astrocyte/RG-like cells showed high velocity toward a neuron fate. As such, this data supports findings of previous research [33,34,35] and contributes to the different methodological approaches fueling the debate on the existence of human neurogenesis [40].

## 8. Distinguishing Astrocytes and Neural Stem Cells (NSCs) in the SVZ and DG

The use of sc-and sn-RNAseq has allowed researchers to transcriptionally distinguish populations of cells involved in adult neurogenesis. This has been especially important in the separation of astrocytes and NSCs within the SVZ and DG. In 2018, Zywitza et al. [8] were the first to distinguish between astrocytes and qNSCs in the SVZ based on the lack of Aqp4 expression in qNSCs. Moreover, qNSCs and aNSCs were resolved on the basis of Ascl1. More transcriptional differences between the astrocytes and NSC populations in the SVZ were subsequently identified by Borrett et al. [15,20] in their analysis of whole live cells from Emx1::Cre;R26^EYFP^ mice and Nkx2.1::Cre-R26^EYFP^ mice. They proposed a considerable list of genes that can be examined together in order to distinguish between these populations, such as the use of Nestin, Dbi, Thbs4, Meg3, Vim for identification of qNSCs, and Aqp4, Agt, S100b, Hbegf, Htra1 for astrocytes. Similar genes were also identified in Redmond et al. [19] to be differentially expressed between astrocyte and qNSC clusters from the analysis of sorted whole cells and single nuclei from hGFAP^GFP^ mice. These studies are in contrast to the work of others in the SVZ that did not find separate clusters for the astrocytes and NSCs [5,6,11,12,13]; or alternatively, developed rescue strategies to separate astrocytes and qNSCs on the basis of a Thbs4 and CD9 likelihood ratio [8].

Separating astrocytes from NSCs in the DG was shown in some, but not all studies. Hochgerner et al. [26] was the first to separate qNSCs and astrocytes in cells isolated from the DG of the hGFAP^GFP^ reporter mice based on their transcriptional differences. Additionally, Batiuk et al. [29] sorted Atp1b2+ cells from the adult C57BL/6J mouse hippocampus to study regional astrocyte heterogeneity, which allowed for the separation between the astrocytes and aNSCs. Alternatively, some transgenic models using CGD, Sox2 or Nestin promoters allow for the isolation of qNSCs in the absence of astrocytes as demonstrated in sc-RNAseq SVZ studies [9,17]. These studies highlight how the commonly used approach of presorting cells using either transgenic mice or immunofluorescent markers can aid in separating a similar population of cells.

The four primate datasets from the DG [31,36] were unable to separate astrocytes from qNSC, as has been achieved in a DG mouse dataset [26], which may in part be due to not presorting the cells. One dataset using the monkey failed to detect clusters with an astrocyte signature [31], whereas the first dataset generated from human hippocampus identified two clusters of astrocytes and one cluster of NSCs, which were later identified as ependymal and not NSCs upon reanalysis by other researchers [35]. The integrated mouse, pig, monkey and human datasets were able to resolve aNSCs and astrocytes [36], but did not provide resolution to differentiate astrocytes from the qNSCs. The astrocyte clusters in this dataset also had high heterogeneity in transcriptional signatures and RNA velocity, thus additional analysis pipelines, or additional studies enriching for subpopulations are ripe for use in future exploration in primate samples.

## 9. Distinguishing Neural Stem and Progenitor Cells (NSCs and NPCs) in the SVZ and DG

Distinguishing qNSCs, aNSCs, and NPCs has been less of a challenge for samples obtained from the SVZ when compared the hippocampus. In the adult mouse SVZ, NPCs and neuroblasts have strong transcriptional signatures that make them easy to separate from NSCs, as demonstrated in many of the reviewed publications e.g., [11,12,38]. Priming of NSCs from a quiescent state has been extensively described in Borrett et al. [15], who showed decreased expression of Mt1, Glul, Cst3 and increased expression of Hmgb3, H2afz and other proliferation-related genes in primed NSCs during this process. Upon activation, aNSCs acquire expression of Ascl1 and Egfr as they move toward NPC or neuroblast fate where they start to show high expression of pro-neural Sox11 and Dlx genes [6]. In addition, signatures of stemness, such as glycolysis pathway activity and lipid metabolism, decrease while providing room for increased ribogenesis and neuronal differentiation, both peaking at the NPC stage [8]. These signatures can be used together with pseudotemporal ordering of cell neurogenesis to truly distinguish NSCs from NPCs.

It has been more difficult to distinguish NSCs and NPCs in the adult mouse DG. Shin et al. [21] were the first to perform sc-RNAseq and provide a comprehensive examination of NSC transitional states using a Nestin^CFP^ reporter mouse. The Waterfall analysis method they developed showed a distinction between qNSCs, aNSCs and NPCs based on the expression of Aldoc, Hopx, and Stmn1, with the former two declining as the NSCs become primed and activated. Artegiani et al. [25] did not corroborate this finding using naïve and Nestin^GFP^ mice, as they did not find many NSCs that were activated and expressing proliferation markers. However, they identified two stages of NPCs using Waterfall, early and late, which differed in their expression of neural fate choice markers. In comparison, Habib et al. [22] used a procedure called Div-Seq to study proliferating cells in the adult DG in a more isolated and controlled fashion. They combined sn-RNAseq with an EdU pulse and sorted the EdU+ cells at different chase times. Although they did not show the clustering of EdU-labeled cells by identity and, therefore, never confirmed the separation between aNSCs and NPCs, the expression pattern of Sox9 and Notch1 in their dataset is clearly distributed along the cell cycle/differentiation continuum. This suggests that the technique has the potential to separate aNSCs from NPCs by varying the EdU injection pulses and chases, and in the future this can be achieved by coupling the procedure to FACS analysis of cell cycle via DNA content. Some of this disagreement among the studies may be due to differences in defining primed NSCs versus aNSCs versus NPCs based on gene expression. The difficulty in reaching consensus reflects the dispersed and heterogeneous nature of hippocampal NSCs and NPCs that is difficult to assess, even with such high-resolution analysis as sc-RNAseq. Therefore, combining sc-RNAseq with transgenic models, as well as other cell labeling methods (e.g., EdU, immunolabeling), may aid in the resolution of qNSCs, aNSCs, and NPCs.

## 10. Identification of the Hypothalamic Tanycytes

Literature examining neurogenesis in the hypothalamic area is growing, and includes two sc-RNAseq [1,2] studies, and one sn-RNAseq [3] study, that have all identified the hypothalamic NSCs called tanycytes. Chen et al. [1] described hypothalamic diversity in adult mice detailing 34 glutamatergic and GABAergic neuronal and 11 non-neuronal sub-types. The study had a high level of transcriptional resolution and identified two types of hypothalamic stem cells: Rax+ tanycytes that were also positive for vimentin and nestin, and Ccdc153+ ependymocytes. This study also provides a few genes that can be used to identify tanycytes, such as Col23a1, Slc16a2, Lhx2, and Ptn, as well as specific markers to be able to distinguish subtypes of tanycytes based on their dorsoventral position. This data was nicely complemented by Kim et al. [2], who used pseudotime and RNA velocity algorithms to describe the developmental trajectory of tanycytes, from early embryo to adult. Using SCENIC to examine the spatial regulators of patterning of hypothalamic subregions, they identified that the expansion and patterning of tanycytes and progenitors in the ventral hypothalamus was governed by Nkx2, and thus may be of interest for examination in adult hypothamic neurogenesis. Hajdarovic et al. [3] further described the regulation of the tanycytes during aging using differential gene expression, SCENIC and pseudotime analysis in single nuclei to determine that tanycytes are regulated by FoxO factors. They also show that gene expression is more stable with age in tanycytes compared to astrocytes, neurons, or oligodendrocytes, yet is also sexually dimorphic. Unlike the studies in the SVZ and dentate, these studies were able to separate the astrocytes from NSCs; however, in the hypothalamus there is overlap in the gene expression between endocytes, ependymal cells, and tanycytes. Albeit few in number, overall these comprehensive studies have helped classify the stem cells in the hypothalamus and provide novel leads to determine their functional dynamics and how they generate mature hypothalamic cells.

## 11. The Transcriptional Dynamics of Upregulation of the Neurogenic Program in the SVZ

The sc-and sn-RNAseq studies in the SVZ have provided a consensus for the pseudotemporal ordering of genes involved in the qNSC and aNSC transition, despite the different cell isolation, various RNAseq protocols, and varying cell cluster resolution. The studies have highlighted important steps in the NSC activation, which include the upregulation of translation and ribogenesis, and the downregulation of astrocytic/stemness genes upon transition from qNSCs to early aNSCs state, followed by cell cycle and proliferation-related gene upregulation during transition from the early to mid-late aNSC stage [5,8]. Lastly, upon transition to later aNSC stages, neuronal differentiation genes become upregulated and astrocyte/NSC identity genes are suppressed. The genetic program of NSC activation and differentiation is governed by Clu, Ccnd2, Dlx2, Dcx, and other genes, including the earlier described Ascl1, all defining distinct molecular signatures of qNSC-like, aNSC-early, aNSC-mid, aNSC-late, and NPC-like populations [5]. This advancement was made due to analysis pipelines like Monocle and its Pseudotime function, which were paramount for lineage inference and pseudotemporal ordering of cell processes across the neurogenic continuum e.g., [8,9,23]. These were also crucial for the discovery of novel regulators of SVZ neurogenesis, such as Troy, which is important in NSC activation and proliferation [6], Lrig1 [16], and Notum, which controls NSC activity and division together with niche occupancy sensing [13].

Unlike the consensus pseudotemporal ordering of genes during the qNSC/aNSC transition, there has been disagreement about which population of cells first upregulates the transcriptional neurogenic program. For instance, Llorens-Bobadilla et al. [4] performed a series of experiments that captured the transcriptomes of Glast ^+^ Prom1^+^ NSCs and Glast^−^Prom1^−^Egfr^+^ NPCs. This analysis unearthed a primed qNSC population, while also discovering evidence of a differentiation program at the aNSC stage. In contrast, Dulken et al. [5] found that late aNSCs lacked the expression of neuronal specification markers like Dcx, Nrxn3, Sp8, etc., which allowed for their separation from NPCs. This study also compared in vitro neurospheres with in vivo FACS-ed aNSCs and suggested the neurospheres separated from their in vivo counterparts due to increased signatures of inflammation and a lack of expression of differentiation markers in neurospheres relative to aNSCS/NPCs. These findings align with previous literature that identified inherent differences between in vivo and in vitro stem and progenitor cell behavior [47]. Additionally, this work supports the proposition that in vivo aNSCs may possess some level of expression of differentiation markers, even if it is less prominent compared to in vivo NPCs.

## 12. The Transcriptional Dynamics Defining Hippocampal Neurogenesis

Shin et al. [21] provided the first sc-RNAseq analysis of adult DG using the Nestin^CFPnuc^ reporter mice to describe the transcriptional mechanisms of adult NSC activation and neurogenesis using their analysis pipeline Waterfall, which incorporated pseudotime and gene expression analyses. This study was paramount in that it did not support the notion of qNSCs as a passive and dormant cell niche. Instead, adult qNSCs continuously integrate various signals from their microenvironment using functional receptors and signaling pathway activity. Upon activation of NSCs, a metabolic shift occurs in qNSCs from lipid and glutathione metabolism (e.g., Spot14, Ascl3 and 6, Acsbg1) as well as glycolysis (aldolase A and C, Ldhb), to oxidative phosphorylation and upregulated expression of mitochondrial genes in NSCs undergoing cell cycle entry. Other hallmarks of NSC activation and proliferation involved transcripts either related to or directly regulating cell cycle, DNA replication, and spindle formation. During the transition between G0 to G1, a marked activation of protein synthesis was observed that also preceded each cell cycle stage transition. These findings have been replicated by others, although all these studies differ in their cluster labels. For example, Hochgerner et al. [26] had similar findings, although they report the qNSCs, aNSCs, and NPCs as individual clusters. Artegiani et al. [25] also report similar pathways of activation in the clusters of qNSCs and early and late NPCs.

Various additional methods were used by others on datasets from the DG, which allowed for a more in-depth evaluation of the transcriptional temporal dynamics occurring in hippocampal neurogenesis. Bergen et al. [28] used the dataset in Hochgerner et al. [26] to demonstrate that developmental trajectories can be inferred using RNA velocity information, which results in determining the fate of the cell seconds to hours ahead. In addition, Zhang and Zhang [30] developed CellPath, a tool that uses RNA velocity data to construct smaller accurate trajectories, or paths, which was shown to be better than the commonly used Pseudotime and Slingshot. This analysis showed that astrocytes arise from qNSCs as well as the resolved novel regulators of neurogenesis, Camk2a and Rasl10a. They have also found a trajectory resembling de-differentiation of adult granule cells, which may involve the expression of Tmsb10. Among other tools that have been recently optimized for trajectory inference in the hippocampus is the VeTra tool, which has shown a promising resolution of lineages compared to Slingshot and other methods [32]. Overall, the abundance of current methods has defined important aspects of dynamic signal integration by qNSCs, as well as revealed the different trajectories of the subpopulations of NSCs and NPCs.

## 13. Effects of Aging and Injury on Adult Neurogenesis

The vast majority of the reviewed studies examine neurogenesis in a naïve state and the use of sc- and sn-RNAseq has only begun to be used to define the transcriptional changes and mechanisms that induce or sustain the reduction or increase in adult neurogenesis in physiological and pathological conditions. For example, four studies have used sc-RNAseq to address how aging reduces production of new cells in the SVZ [7,10,12,17], two studies have been completed in the hippocampus [25,37], and one in the hypothalamus [3]. Shi et al. [7], using sc-RNAseq analysis, demonstrated that, overall, aging reduced cell proliferation, altered cell cycle regulation, and induced the inflammatory SVZ microenvironment. This is consistent with the findings in the dentate that aged adult mice had a smaller ratio of NSCs to dividing NPCs [25]. Quiescent NSCs from the aging SVZ were further shown by Kalamakis et al. [10] to be more resistant to becoming aNSCs, with an accompanying increase in the qNSC population to preserve the small pool of NSCs. The proposed mechanism for the reduction in transition from qNSC to aNCSs was hypothesized by Dulken et al. [12] due to the infiltration by immune cells and increased gamma-interferon signaling in the aging SVZ. Most recently, Xie et al. [17] additionally explored the role of aging utilizing a CGD^GFP^ reporter mouse and found seven distinguishable subgroups of NSCs and NPCs. Using this higher level of specificity to separate their cell populations, they confirm that aging is associated with the failure of qNSC to transition to progenitors. They additionally show the transcriptional down-regulation of the cell cycle, protein and macromolecular catabolism pathways, as well as master regulator genes, such as Myc, Sp1, Srebf2, E2f1. Taken together, these studies have opened up new avenues for future data mining to identify more mechanisms by which the balance between quiescence and activation of NSCs is perturbed in aging.

The use of sc-RNAseq in the SVZ has also addressed pertinent questions relating to the increase in NSCs and NPCs within the SVZ niche to various types of injury. The first sc-RNAseq study of adult SVZ cells which described the local neurogenic niche and its transcriptional response to ischemic injury in adult mice [4] showed that ischemic injury triggers the entry of qNSCs into a state primed for activation via gamma-interferon signaling. Similarly, mild traumatic brain injury is associated with the entry of resident astrocytes into a neurogenic program and the subsequent expansion of the qNSC and aNSC pools [18]. Importantly, in this study there was direct evidence of differentiation of aNSCs into neuroblasts, supporting the notion that aNSCs possess differentiation signals to commit to neurogenesis.

## 14. The Future

Single-cell and -nucleus RNA sequencing has significantly advanced our understanding of the molecular features and dynamics of stem cells housed in the adult murine SVZ, DG, and hypothalamus. There is no doubt that continuous use of this technology will make it more widely available to unravel novel regulators and pathways that allow for the generation of neurons from NSCs in the adult brain. Given the rapidly evolving abundance of new pipelines and analyses being continuously generated, the future is also bright with regard to resolving specific trajectories of development and transcriptional networks of different cellular subtypes and allowing for the optimal resolution of cells with similar transcriptional signatures. For instance, with the increased availability of sc-RNAseq data, neural networks can be used for cell type assignment [48] and for uncovering complex gene networks [49], which will enhance our interpretation of these datasets. Furthermore, it is exciting to imagine how combining sc-RNAseq with chromatin analysis in scATACseq [50], glycan analysis in scGRseq [51], or spatial sc-RNAseq technologies [52] will provide more high-throughput information from individual cells and more depth to our understanding of how neurons are generated in the adult brain.

## Figures and Tables

**Table 1 cells-11-01633-t001:** Primary literature using sc-RNAseq or sn-RNAseq to assessing adult neurogenesis in the SVZ.

Author (Year)	Species	Age	Sex	Models	Method	Platform	Analysis	Number of Cells
Llorens-Bobadilla et al. (2015) [4]	Mouse	8–12 wk	M	C57BL/6 mice	SVZ wholemount digested with trypsin, whole cells sorted and frozen	Smart-seq2	FactoMineR, Monocle, likelihood-ratio	<100–1000
Dulken et al. (2017) [5]	Mouse	3 mo	M	GFAP^GFP^ reporter mice, datasets Llorens-Bobadilla et al. (2015) & Shin et al. (2015)	SVZ microdissected, digested with papain, whole cells sorted and processed live	Fluidigm C1	GBM modeling, Monocle, SCDE, GSEA	329
Basak et al. (2018) [6]	Mouse	2 d–1 yr	F	Troy^GFPiresCreER^ and Ki67^RFP^ reporter mice	SVZ wholemount enzymatically digested, whole cells sorted and processed live	CEL-seq	RaceID2, Descan, Pseudotime in TSCAN	1465
Shi et al. (2017) [7]	Mouse	1, 24 mo	F + M	C57BL/6 mice, dataset Llorens-Bobadilla et al. (2015)	SVZ microdissected, digested with papain, cultured as neurospheres, whole cells sorted and frozen	Smart-Seq2	t-SNE, WGCNA, DESeq2, GO	22
Zywitza et al. (2018) [8]	Mouse	2–4 mo	M, F, F + M	C57BL/6N mice Lrp2 KO mice, dataset Artegiani et al. (2017)	SVZ microdissected digested with papain, whole cells processed live or methanol-fixed	Drop-seq	Seurat, SNN-cliq, Velocyto	9804
Shah et al. (2018) [9]	Mouse	2–6 mo	F + M	aSMA::CreER^T2^; R26^tdTomato^/Sox2^GFP^ mice	SVZ microdissected, digested with papain, whole cells sorted and processed live	10× Genomics	Seurat, SCDE	1200, 6000
Kalamakis et al. (2019) [10]	Mouse	2, 22, 23 mo	M	C57BL/6J mice and dataset from Llorens-Bobadilla et al. (2015)	SVZ wholemount digested with trypsin, whole cells sorted and frozen	Smart-Seq2	Seurat, Monocle, clusterProfile, DESeq2	>2000
Mizrak et al. (2019) [11]	Mouse	8–10 wk	M, F	hGFAP::CreER^T^;R26^tdTomato^ mice and datasets from Llorens-Bobadilla et al. (2015), Dulken et al. (2017)	Lateral and septal SVZ wholemounts digested with papain, whole cells processed live	Drop-seq	Phenograph, GSEA, SCDE	41,000
Dulken et al. (2019) [12]	Mouse	3, 28–29 mo	M	C57BL/6NIA mice	SVZ microdissected, digested with papain, whole cells sorted and processed live	10× Genomics	Seurat, Enrichr	14,685
Mizrak et al. (2020) [13]	Mouse	8–10 wk	M	hGFAP::CreER^T^;R26^tdTomato^ mice ratNes::FLPOER;R26^TdTomato^ mice	Lateral and septal SVZ wholemounts digested with papain, whole cells processed live	Drop-seq	Phenograph, SCDE	56,000
Magnusson et al. (2020) [14]	Mouse	>2 mo	M, F	Cx3::CreER;Rbpj^fl/fl^;R26^tdTomato/YFP^ mice, AAV-Cre injection into Rbpj^fl/fl^;R26^tdTomato^ mice, datasets from Zywitza et al. (2018), and Hochgerner et al. (2018)	Microdissected striatum digested with papain, whole cells sorted and frozen	Smart-Seq2, 10× Genomics	Seurat, Monocle	1393 203
Borrett et al. (2020) [15]	Mouse	E–2 mo	F + M	Emx1::Cre;R26^EYFP^ mice Nkx2.1::Cre;R26^EYFP^ mice	Dorsal and lateral SVZ microdissected, digested enzymatically, whole cells sorted and processed live	10× Genomics	Seurat	>6000
Nam and Capecchi (2020) [16]				Mizrak et al. (2020) dataset			Seurat	
Xie et al. (2020) [17]	Mouse	2 wo–15 mo	F + M	CGD^GFP^ reporter mice, datasets from Dulken et al. (2017), Llorens-Bobadilla et al. (2015), Codega et al. (2014)	Wholemount SVZ digested with papain, whole cells sorted and processed live	Drop-seq	Seurat, Pseudotime, TFactS, String	5600
Chen et al. (2021) [18]	Mouse	8–10 wo	M	C57BL/6J mice	Microdissected SVZ frozen, homogenized, and nuclei processed after sucrose gradient centrifugation	10× Genomics	Seurat, GO, CellPhoneDB, Monocle	15,754
Cebrian-Silla et al. (2021) [19]	Mouse	4–5 wo	M, F	hGFAP^GFP^ reporter mice	Microdissected SVZ digested with papain, whole cells multiplexed, processed live	10× Genomics	Seurat, scVelo, GO	30,897
				CD1-elite mice	Anterior/posterior-dorsal/ventral SVZ microdissected SVZ frozen, homogenized, and nuclei processed after sucrose gradient centrifugation			45,820
Borrett et al. (2022) [20]				Datasets from Hochgerner et al. (2018), Borrett et al. (2020)			Seurat, Monocle, GSEA	

Abbreviations: F = female M = male, F + M = female and male samples pooled; nr= sex not reported; E = embryonic; d = days; w = weeks, mo = months; yr = years.

**Table 2 cells-11-01633-t002:** Primary literature using sc-RNAseq or sn-RNAseq to assessing adult neurogenesis in the hippocampus.

Author (Year)	Species	Age	Sex	Models	Method	Platform	Analysis	Number of Cells
Shin et al. (2015) [21]	Mouse	8–12 wk	M	Nestin^CFPnuc^ reporter mice	Microdissected DG digested with papain, whole live cells sorted and frozen	Smart-seq2	Waterfall	<200
Habib et al. (2016) [22]	Mouse	<2 yr	M	AAV1/2 injection into vGAT-Cre mice	Microdissected DG and other hippocampal subregions digested, fixed, sorted, nuclei frozen and processed with “Frankenstein” method [23]	Smart-seq2	Seurat	1367
Habib et al. (2017) [24]	Mouse, Human	10–14 wk mice, 40–65 yr human	M	C57Bl/6 mice and humans	Frozen hippocampus dissected and nuclei were processed using the sucrose gradient centrifugation or the “Frankenstein” method [23]	Drop-seq, DroNc-seq	Seurat	Mouse: 13,313 Human: 14,963
Artegiani et al. (2017) [25]	Mouse	6 & 10 wk, >1 yr	F+M	C57BL/6 mice Nestin^GFP^ reporter mice	Microdissected DG digested with papain, whole cells sorted and frozen	SORT-seq	RaceID2, StemID, Waterfall	1408
Hochgerner et al. (2018) [26]	Mouse	2–5 wk	F+M	C57BL/6 mice hGAFP^GFP^ reporter mice	Microdissected DG digested with papain, sorted, whole cells sorted and processed live	Fluidigm C1, 10× Genomics	Matlab	24,185
Lisi et al. (2019) [27]	Mouse	4–6 wk, 32–40 wk	F	rTg4510 tauopathy mouse model	Dissected hippocampus digested with papain, whole cells processed live	Drop-seq	Seurat, Monocle	>3000
Bergen et al. (2020) [28]				Dataset from Hochgerner et al. (2018)			scVelo	
Batiuk et al. (2020) [29]	Mouse	2 mo	F+M	C57BL/6J mice	Dissected hippocampus digested with papain, whole cells sorted and frozen	Smart-seq2	Seurat, GO	2015
Zhang and Zhang (2021) [30]				Dataset from Hochgerner et al. (2018)			scVelo, Velocyto, CellPath, Slingshot, Vdpt, reCAT	
Zhang et al. (2021) [31]	Macaque	4–6 yr, 18–21 yr	M, F	Cynomolgus macaques	Frozen hippocampus is homogenized, nuclei sorted, and processed	10× Genomics	Seurat, Monocle, GO, SCENIC, CellPhoneDB, Pseudotime	>8000
Weng et al. (2021) [32]				Dataset from Hochgerner et al. (2018)			VeTra	
Tran et al. (2021) [33]	Human	40–69 yr	M	Human	Frozen hippocampus dissected and nuclei were processed using the “Frankenstein” method [23]	10× Genomics	Bioconductor, MAGMA	70,615
Ayhan et al. (2021) [34]	Human	24–60 yr	M, F	Human and datasets from Habib et al. (2016, 2017), and Batiuk et al. (2020)	Frozen anterior and posterior hippocampus dissected and nuclei were processed using the “Frankenstein” method [23]	10× Genomics	Seurat, GO, MAST	129,908
Sorrells et al. (2021) [35]				Dataset from Habib et al. (2017)			Seurat	
Franjic et al. (2022) [36]	Human, Macaque, Pig	48–58 yr human 8–14 yr macaque 3 mo pig	M, F	Human, rhesus macaques, pig, and datasets from Ayhan et al. (2021), Hochgerner et al. (2018) and Zhong et al. (2020)	Frozen DG and other hippocampal regions microdissected, homogenized, nuclei processed after sucrose gradient centrifugation	10× Genomics	Seurat, Velocyto, scVelo	Human: 139,187 Macaque: 36,107 Pig: 38,851
Borrett et al. (2022) [20]				Datasets from Hochgerner et al. (2018) and Borrett et al. (2020)			Seurat, Monocle, GSEA	
Schneider et al. (2022) [37]	Mouse	2–14 mo	F+M	hGAFP^eGFP^ reporter mice	Microdissected DG digested with trypsin, whole cells processed live	10× Genomics	Seurat, GO	>5000

Abbreviations: F = female M = male, F + M = female and male samples pooled; nr = sex not reported; E = embryonic; d = days; w = weeks, mo = months; yr = years.

## Data Availability

Not applicable.

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
