# Peer review of "Single-Cell and Single-Nucleus RNAseq Analysis of Adult Neurogenesis"

_cells, 2022, doi:10.3390/cells11101633_

Round 1

Reviewer 1 Report

In this work by Kalinina and Lagace, the authors review the available literature of single-cell and single-nucleus RNAseq transcriptomics analysis of adult neurogenesis in the subventricular zone, the dentate gyrus of the hippocampus, and the hypothalamus. The manuscript is well organized and provides condensed information on the current status of adult neurogenesis from the single-cell perspective.

The work is highly pertinent and of great interest to the field, and the increasing amount of published works to date demand careful examination. The authors have divided their analysis into 15 different sections that allow the reader to get a full picture in an easy manner. The authors are to be commended by their appropriate summary of methodological variations (species demographics, sample collection, plate- Vs droplet-based methods, number of cells, and analysis pipelines) prior to dissecting out the neurogenic lineages for each niche. The authors then focused their attention on the differences and similarities between astrocytes and NSCs in both the SVZ and DG, and next did similarly between NSCs and NPCs but only in the DG (not in the SVZ). This reviewer feels that adding one paragraph in section 10 about NSCs-NPCs in the SVZ would complete such gap. It would be good if the authors could come up with a consensus definition of a transcriptional signature for each cell type belonging to the neurogenic lineage.

Regarding the controversy on adult neurogenesis in humans, the work by Sorrells et al, J Neurosci, 2021 (PMID: 33762407) is highly relevant and deserves to be discussed here as well as they analyzed other datasets that could be also included in this review (PMID: 25700174, PMID: 30154505, PMID: 28846088). Importantly, Franjic and cols. found that DCX-expressing cells in the human DG are surprisingly inhibitory - not excitatory. On another note, recent work has also shown the presence of immature excitatory neuroblasts in the adolescent human amygdala with the help of single-nucleus RNAseq (PMID: 31227709). I am well aware that the amygdala is out of the scope of this review but it could be useful for the authors to support the notion that immature neurons in the adult human brain seem to be generated during perinatal stages.

Minor points

  1. "Franjic et al, Neuron, 2021" also presented mouse data (Table 2).
  2. Please cite the final version of "Cebrián-Silla et al, eLife, 2021" instead of "Redmond et al, BioRxiv, 2021". Same for reference number 17 (Habib et al, Nat Methods, 2017).
  3. For work performed on C57BL/6 mice, could the authors please check the specific substrain (C57BL/6J or C57BL/6N) if the information is provided initially? It is important to clarify this issue as several differences have emerged in recent years between substrains.
  4. Although it is well written in a very clear and understandable way, there are however several typos that need to be corrected. Here are some that I found:
    Line (L) 94: "future,"
    L100-101: space before commas
    L136: "florescent"
    L144: "have not be used"
    L302: "was the first to separated"
    L307: "promoter do not labels"
    L339: "FACs"
    L427: "Various others methods"

Author Response

We appreciate your time and comprehensive review of our article.  Please find our responses in red below. 

The authors then focused their attention on the differences and similarities between astrocytes and NSCs in both the SVZ and DG, and next did similarly between NSCs and NPCs but only in the DG (not in the SVZ). This reviewer feels that adding one paragraph in section 10 about NSCs-NPCs in the SVZ would complete such gap. It would be good if the authors could come up with a consensus definition of a transcriptional signature for each cell type belonging to the neurogenic lineage.

We agree with this suggestion. We have added in a new paragraph in Section 10 on the NSC-NPCs in the SVZ that includes a consensus definition for their transcriptional signatures.

Regarding the controversy on adult neurogenesis in humans, the work by Sorrells et al, J Neurosci, 2021 (PMID: 33762407) is highly relevant and deserves to be discussed here as well as they analyzed other datasets that could be also included in this review (PMID: 25700174, PMID: 30154505, PMID: 28846088).

We appreciate the suggestion to add these additional references and have done the following:

Sorrells et al, J Neurosci, 2021 (PMID: 33762407) is a comprehensive dual perspective article on the debate on adult neurogenesis that has reanalyzed the dataset found in Habib et al 2017 in Figure 5. We have included this citation in the table and when appropriate within the text.

Habib et al, Nat Methods 2017 (PMID 33762407) completed snRNAseq with DroNc-seq in the human and mouse hippocampus. We have included this citation in the table and when appropriate within the text.

Based on this reviewers comments, we have also added in the papers by Tran et al. 2021 (PMID: 34582785) and Ayhan et al. (34582785) since they also perform snRNA-seq analysis in human DG hippocampus.

These additionally papers are also summarized in a new paragraph in the section called “Identification of Cell Types within the Dentate”, in order to provide a more concise summary of the findings from the studies using human samples and snRNAseq. 

Importantly, Franjic and cols. found that DCX-expressing cells in the human DG are surprisingly inhibitory - not excitatory.

We agree that this is a very important and surprising.  We have not include this information in this review, since we have not provided that level of depth about other cells types identified through complementary IHC analysis, and feel it is beyond the scope of this review.

On another note, recent work has also shown the presence of immature excitatory neuroblasts in the adolescent human amygdala with the help of single-nucleus RNAseq (PMID: 31227709). I am well aware that the amygdala is out of the scope of this review but it could be useful for the authors to support the notion that immature neurons in the adult human brain seem to be generated during perinatal stages.

This is another interesting paper, but as acknowledged by the reviewer is beyond the scope of this paper and thus was not included in the revised paper to keep the paper focused to our areas of interest.

"Franjic et al, Neuron, 2021" also presented mouse data (Table 2).

In Table 2, we have listed under the Model column the sources of the cells utilized, or referenced the data that was used from a previous publication.

For Franjic et al, the mouse data came from Hochgerner et al 2018, therefore we included “Dataset from Hochgerner et al. (2018)” in the Model column.

We had also highlighted in the Methodology: Species Demographics text that for Franjic et al, “The paper also provides cross-species comparisons between the data they generated and a previously published mouse dataset [6] that reveal species-specific differences”.

Please cite the final version of "Cebrián-Silla et al, eLife, 2021" instead of "Redmond et al, BioRxiv, 2021".

Thank you, we updated the Cebrian-Silla et al (2021) reference.

Same for reference number 17 (Habib et al, Nat Methods, 2017).

We added the reference for Habib et al (2017) and also keep the Habib et al (2016) publication we cited in our review originally.

For work performed on C57BL/6 mice, could the authors please check the specific substrain (C57BL/6J or C57BL/6N) if the information is provided initially? It is important to clarify this issue as several differences have emerged in recent years between substrains.

Great suggestion and we agree this is important. When we could find the substrain information in the publication or through contact with the authors, we have added this information to the table for all studies using C57 mice.

Although it is well written in a very clear and understandable way, there are however several typos that need to be corrected. Here are some that I found:

   Line (L) 94: "future,"

   L100-101: space before commas

   L136: "florescent"

   L144: "have not be used"

   L302: "was the first to separated"

   L307: "promoter do not labels"

   L339: "FACs"

   L427: "Various others methods"

Thank you, these have been fixed and the paper was reviewed for additional typos.

Reviewer 2 Report

The manuscript entitled Single-Cell and Single-Nucleus RNAseq Analysis of Adult Neurogenesis is a narrative review about studies mainly carried out in mice about transcriptional analyses of neural stem cells in brain neurogenetic regions.

The manuscript is interesting and clearly written.

A few suggestions:

In Table 1 and Table 2 references should be numbered according to the reference list in the main text to make them easier to find.

Since the information about animal sex is important, the authors should try to obtain it by writ to corresponding authors of the respective studies to ask for this piece of data in order to include it in the manuscript.

There are a few typos at lines 51, 134, 302, 307, 311, 323, 347, 417 that should be corrected.

Author Response

In Table 1 and Table 2 references should be numbered according to the reference list in the main text to make them easier to find.

Thank you for this suggestion, we have indexed the references in the table as per the reference section.

Since the information about animal sex is important, the authors should try to obtain it by writ to corresponding authors of the respective studies to ask for this piece of data in order to include it in the manuscript.

We contacted all the corresponding authors and added the animal sex information for all papers.

There are a few typos at lines 51, 134, 302, 307, 311, 323, 347, 417 that should be corrected.

Thank you. The typos have been fixed.

Round 2

Reviewer 1 Report

The authors have addressed all my concerns and answered my questions. This review is relevant and will be of interest for the field.